# Phylogenetic Network Analyses Reveal the Influence of Transmission Clustering on the Spread of HIV Drug Resistance in Quebec from 2002 to 2022

**DOI:** 10.3390/v16081230

**Published:** 2024-07-31

**Authors:** Bluma G. Brenner, Ruxandra-Ilinca Ibanescu, Maureen Oliveira, Guillaume Margaillan, Bertrand Lebouché, Réjean Thomas, Jean Guy Baril, René-Pierre Lorgeoux, Michel Roger, Jean-Pierre Routy

**Affiliations:** 1McGill University Centre for Viral Diseases, Lady Davis Institute for Medical Research, Montreal, QC H3T 1E2, Canada; ribanescu@jgh.mcgill.ca (R.-I.I.); maureen.oliveira@jgh.mcgill.ca (M.O.); 2Department of Microbiology and Immunology, McGill University, Montreal, QC H3A 2B4, Canada; 3Department of Medicine (Surgery, Infectious Disease), McGill University, Montreal, QC H3A 0G4, Canada; 4Département de Microbiologie et d’Immunologie et Centre de Recherche du Centre Hospitalier de l’Université de Montréal (CHUM), Montreal, QC H2X 0C1, Canada; guillaume.margaillon.chum@ssss.gouv.qc.ca; 5Chronic Viral Illness Service, McGill University Health Centre, Montreal, QC H4A 3J1, Canada; bertrand.lebouche@mcgill.ca (B.L.); jean-pierre.routy@mcgill.ca (J.-P.R.); 6Clinique Médicale l’Actuel, Montreal, QC H2L 4P9, Canada; rejean.thomas@lactuel.ca; 7Clinique Médicale du Quartier Latin, Montreal, QC H2L 4E9, Canada; jgbaril@videtron.ca; 8Gilead Sciences Inc., Montreal, QC L5N 7K2, Canada; rene-pierre.lorgeoux@gilead.com; 9Primary HIV Infection (PHI) Cohort, Centre de Recherche du CHUM, Montreal, QC H2X 0A9, Canada; michel.roger.chum@ssss.gouv.qc.ca

**Keywords:** HIV-1, acquired and transmitted drug resistance, antiretroviral therapy, phylogenetic clustering, M184V

## Abstract

Background: HIV drug resistance (HIV-DR) may jeopardize the benefit of antiretroviral therapy (ART) in treatment and prevention. This study utilized viral phylogenetics to resolve the influence of transmission networks on sustaining the spread of HIV-DR in Quebec spanning 2002 to 2022. Methods: Time trends in acquired (ADR) and transmitted drug resistance (TDR) were delineated in treatment-experienced (*n* = 3500) and ART-naïve persons (*n* = 6011) with subtype B infections. Similarly, non-B-subtype HIV-DR networks were assessed pre- (*n* = 1577) and post-ART experience (*n* = 488). Risks of acquisition of resistance-associated mutations (RAMs) were related to clustering using 1, 2–5, vs. 6+ members per cluster as categorical variables. Results: Despite steady declines in treatment failure and ADR since 2007, rates of TDR among newly infected, ART-naive persons remained at 14% spanning the 2007–2011, 2012–2016, and 2017–2022 periods. Notably, half of new infections among men having sex with men and heterosexual groups were linked in large, clustered networks having a median of 35 (14–73 IQR) and 16 (9–26 IQR) members per cluster, respectively. Cluster membership and size were implicated in forward transmission of non-nucleoside reverse transcriptase inhibitor NNRTI RAMs (9%) and thymidine analogue mutations (TAMs) (5%). In contrast, transmission of M184V, K65R, and integrase inhibitors (1–2%) remained rare. Levels of TDR reflected viral replicative fitness. The median baseline viremia in ART-naïve groups having no RAMs, NNRTI RAMs, TAMs, and M184VI were 46.088, 38,447, 20,330, and 6811 copies/mL, respectively (*p* < 0.0001). Conclusion: Phylogenetics emphasize the need to prioritize ART and pre-exposure prophylaxis strategies to avert the expansion of transmission cascades of HIV-DR.

## 1. Introduction

Remarkable advances in combination antiretroviral therapy (ART) have transformed HIV-1 from a deadly to a lifelong treatable disease. Beyond the benefit for the individual, ART reduces community viral load and prevents onward spread of HIV at a population level. In 2014, the World Health Organization (WHO)/UNAIDS launched the “95-95-95” treatment-as-prevention initiative to reduce the global incidence of HIV from 2 million cases in 2010 to 500,000 and 200,000 in 2020 and 2030, respectively [1]. All nations were called upon to diagnose 95% of their HIV-infected populations, provide ART to 95% of those infected, and achieve viral suppression in 95% of those treated by 2030. Current HIV prevention guidelines incentivize HIV testing, rapid initiation of ART for all those infected, and expanded access of pre-exposure prophylaxis (PrEP) for HIV-negative persons at high risk for infection, e.g., men having sex with men (MSM). The concerted scale-up of ART to 29.8 million of the 38 million persons living with HIV has contributed to promising 38% declines in heterosexual epidemics across Africa [2,3]

The disturbing increases in rates of acquired drug resistance (ADR) and transmitted drug resistance (TDR) threaten the sustained benefit of ART and PrEP options in pandemic control [4]. Ongoing surveillance studies by the WHO report exponential increases in the prevalence of transmitted resistance to non-nucleoside reverse transcriptase (RT) inhibitors (NNRTIs), frequently exceeding 10% of newly infected ART-naïve populations [2,3,4]. Notably, single point mutations, including K103N, Y181C, and G190A, can render viruses highly resistant to NNRTIs, including efavirenz (EFV) and nevirapine (NVP). To a lesser extent, transmission of resistance to thymidine analogue mutations (TAMs) (i.e., M41L, D67N, K70R, L210W, T215 D/L/S/N revertants, and K219Q) have increased over time despite the discontinued use of zidovudine (AZT) and stavudine (d4T) in clinical practice. In contrast, M184I/V resistance to lamivudine (3TC) or emtricitabine (FTC) acquired upon virologic failure disappears within weeks or months following transmission [5]. 

The rise in drug resistance to NNRTIs has led to transitions to integrase strand transfer inhibitor (INSTI)-anchored regimens, including dolutegravir (DTG), bictegravir (BIC), or cabotegravir (CAB). The updated WHO report in 2024 forewarns growing levels of resistance to dolutegravir [6]. In addition, recent INSTI-based streamlined two-drug, intermittent ART regimens and injectable formulations may be adversely affected by the genesis of transmissible drug-resistant variants [7,8,9].

Our studies have applied viral phylogenetic linkage as a molecular framework to demonstrate the role of transmission networks in the forward spread of HIV among MSM and heterosexual groups in Quebec from 2002 to 2022 [10]. Our findings herein reveal the influence of clustered outbreaks in half of TDR among ART-naïve populations. Pretreatment viral set-points were significant predictors of the differential persistence of viral species bearing NNRTIs, TAMs, and M184V RAMs. TDR may hamper the efficacy and durability of newer ART and PrEP options.

## 2. Materials and Methods

### 2.1. Study Design

The Quebec drug resistance program, operational since 2002, has accrued HIV pol sequences from 11,571 of the estimated 17,726 (15,600–20,300) persons living with HIV in the province. Genotyping was recommended for all newly diagnosed persons, stratified according to physician-designated clinical indication of primary infection (under 6 months) or ART-naïve recent infection (>6 months post-infection). ART-experienced patients are stratified as failing first or subsequent regimens. Test requisition information included sampling date, clinical site, and participant characteristics (age, gender, viral load, and treatment status).

Phylogenetic network analyses, performed using MEGA10-integrated software, version 10 (www.megasoftware.net) (Access date 10 January 2023), followed annual trends in the spread of HIV subtypes, drug resistance, and transmission clustering over the past two decades [10]. Non-nominative subject identifiers assured patient anonymity, using birthdate cross-identifiers to identify replicate sampling. Time trends in HIV drug resistance (HIV-DR) were analyzed in treatment-naïve and ART-experienced populations, sub-stratified according to viral subtype and gender. Phylogenetic analyses assessed the influence of clustering on HIV-DR including:
Transmitted resistance (TDR) in ART-naïve persons acquiring subtype-B infections (*n* = 6011): this group encompassed the predominant men having sex with men (MSM) epidemic (male singletons and male–male clusters, *n* = 4854) and the heterosexual (HET) subtype B epidemic from persons arriving from Haiti, the Caribbean, and Latin and South America (female singletons and mixed gender clusters, 56% female, *n* = 1157). Acquired resistance (ADR) in chronic treated persons with subtype-B infections (*n* = 3500): this group incorporated all genotyped persons failing first or subsequent treatment regimens (*n* = 6013 sequences from 3500 persons). Pretreatment resistance (PDR) and ADR in persons having non-B-subtype infections (*n* = 2065) this group included newcomers to the province arriving from francophone countries in Africa, Europe, and Asia. Overall, 1577 of those genotyped were reported as ART-naive.

### 2.2. Sequence Analyses of Transmission Networks and Drug Resistance

Viral sequences, spanning HIV protease and reverse transcriptase (HXB2 nucleotide positions 2262→3290 or 2253→3749), were aligned to consensus HXB2 sequences, removing gaps, and cut to identical sequence lengths using Bio Edit version 7.7 to address different protocols used at different time periods. Subtype B and non-B-subtype trees were rooted against subtype K consensus sequence [10]. Genotyping spanning the viral integrase (positions 4230→5093) was performed since 2015 [11]. 

Phylogenetic trees were reconstructed using MEGA10-integrated software (www.megasoftware.net). Transmission clustering was assigned based on high bootstrap support (generally > 95%) and short genetic distance (generally TN93 under 1.5% substitutions/site). Members within individual clusters were assessed for shared natural RT/PR polymorphisms and HIV-DR RAMs. Molecular transmission networks were also constructed using MicrobeTrace (http://github.com/cdcgov/microbetrace, accessed on 10 January 2023) at genetic distance thresholds of 1.5% and 2.5% [12]. Coloring of individual nodes visualized the differential spread of HIV-DR mutations. 

Each newly genotyped person was assigned a non-nominative phylogenetic identifier based on HIV-1 subtype, cluster group membership, sex, and disease stage at first presentation. The phylogenetic cluster code assigned at first presentation remained invariant, permitting phylodynamic tracking of cluster evolution (cluster size and growth rate) spanning the 2002 and 2020 periods.

### 2.3. Drug Resistance and Statistical Analyses

HIV-DR was monitored applying the CEDRIC-HIV checklist (http://cedric-hiv.com/wp-content/uploads/2023/08/CEDRIC-HIV_Eng_v2.pdf, accessed on 1 March 2023) [13]. Drug resistance mutations were defined according to the updated World Health Organization 2009 list of mutations for surveillance of TDRs [14,15]. The prevalence of mutations that may contribute to resistance to newer second-generation NNRTIs, rilpivirine (RIL), etravirine (ETV) and doravirine (DOR), are included in figures and tables. These accessory mutations, including E138A/G/Q/R, E138K, L234I, V318F, and N348I, were excluded from statistical analyses of rates of TDR or ADR. 

Time trends in rates of ADR and TDR were monitored annually, as well as cumulatively over the 2007–2011, 2012–2016, and 2017–2022 periods. Cluster membership and size were evaluated as a categorical variable (singletons vs. 2–5 vs. 6+ members/cluster), using Χ^2^ test statistics to compute the significance and odds ratios in the forward spread of individual RAMs. One-way ANOVA analysis (nonparametric) followed by Kruskal Wallis multiple comparisons ascertained the influence of select RAMs on pretreatment viremia and viral transmissibility. Statistical analyses were performed using GraphPad Prism version 10.2.3 software (www.graphpad.com, accessed on 24 July 2024).

## 3. Results

### 3.1. Overall Prevalence of Acquired and Transmitted Drug Resistance in Quebec

This study applied a phylogenetic framework to investigate the influence of transmission clustering on onward spread of HIV-DR in Quebec from 2002 to 2022. Sequence datasets were accrued from 11,571 genotyped individuals, including both ART-naïve and ART-experienced populations. Overall, 14.7% and 13.9% of newly diagnosed, ART-naïve persons bearing subtype B (*n* = 884/6011) or non-B-subtype infections (*n* = 1577) harbored resistance to one or more drug class, respectively (Table 1).

The rates of TDR across drug classes were similar in subtype B and non-B-subtype groups. Resistance to drug classes in subtype B/non-B-subtype groups were 8.8%/9.4% for NNRTIs, 5.9%/6.2% for NRTIs, and 2.4%/2.0% for PIs (Table 1). The most common transmitted RAMs in subtype B were K103N (4.3%), G190A (2.8%), T215 revertants (4.5%), M41L (1.9%), M184V (1.8%), and L90M (1.1%) (Figure 1).

The differential distributions of acquired and transmitted RAMs are depicted in Figure 1. The high transmissibility of K103N and G190A species, conferring resistance to the first-generation NNRTIs, EFV and NVP, reflected their limited impact on viral replicative fitness. In contrast, the rates of ADR and TDR to the next-generation NNRTIs, including doravirine (V108I, V106A/M, F227L, M230L, and Y318F) and rilpivirine (L100I, K101EP, E138K, K101E/P, and M230L) remain rare in the province. Although the prevalence of E138K (0.2%) was rare, the E138A natural polymorphism was present in treatment-naïve persons with subtype B and non-B-subtype infections (2.7% and 5.1%, respectively). The latter mutation has been reported to reduce rilpivirine and etravirine susceptibility by 2-fold [16,17].

To a lesser extent, transmission of resistance to NNRTIs included the thymidine analogue mutations (TAMs) (i.e., M41L, D67N, K70R, L210W, T215 D/L/S/N revertants, and K219Q). The incidence of M184V was low (1.8%) in ART-naïve persons as compared to treatment-experienced persons (42.2%) (Table 1). This reflects the reported fitness costs of M184V that leads to rapid reversion within weeks following transmission [5]. Similarly, acquisition and transmission of K65R remain rare (0.1%) despite the widespread use of TDF/TAF as anchor drugs in ART regimens (Figure 1).

### 3.2. Longitudinal Trends in the Spread of Acquired and Transmitted Resistance

We compared the frequencies of ADR and TDR over the 2007–2011, 2012–2016, and 2017–2022 periods (Figure 2). Over these three periods, numbers of genotyped persons failing ART (VL > 400 copies/mL) (80%→59%→48%) and rates of ADR steadily declined (100→42%→23%). This was concomitant to the transition from NNRTI- to integrase-inhibitor-based regimens (Figure 2A). In stark contrast, rates of transmitted resistance to NNRTIs to newly infected persons remained steady at 9.5%, 8.9%, and 8.8% during these three respective periods (Figure 2B). Similarly, the transmission of TAMs and T215 revertants persisted (5.4%→2.8%→7.0%) over time despite the discontinued use of AZT and d4T in clinical practice (Figure 2B). The transmission of M184V,I in newly diagnosed persons rose marginally in the 2017–2022 period (1.0%→0.8%→3.3%, respectively, Χ^2^ = 33.7, *p* < 0001). 

### 3.3. Influence of Clustered Outbreaks on the Persistence of HIV TDR in Newly Infected Persons

Phylogenetic network analyses were performed to better resolve the influence of HIV-1 transmission clustering on the forward spread of TDR. Individuals with subtype B and non-B-subtype infections were segregated into three risk groups based on sex, gender, and putative cluster group. The predominant subtype B epidemic among the MSM epidemic (65% of cases, 2002–2023) includes male-only singleton transmissions (*n* = 1804), 2–5 clusters (*n* = 1168), and 6+ clusters (*n* = 3039), representing 30%, 17%, and 52% of genotyped infections, respectively. For the subtype B HET epidemic, female singletons, mixed-gender 2–5, and 6+ clusters comprised 30%, 29%, and 41% of infections, respectively.

There were significant influences of clustered outbreaks in the dissemination of TDR in the province. Figure 3 and Figure 4 depict the role of clustered outbreaks in the forward spread of subtype B epidemic bearing HIV-DR mutations (*n* = 1041).

TDR to NNRTIs was frequently associated with large cluster outbreaks (6+ members). This included the circled cluster bearing G190A (C50, *n* = 160 members) and clusters bearing K103N, E138A, and Y181C (Figure 3). For the NRTI drug class, TAMs and M1184V were more likely to be present as isolated singleton transmissions rather than belonging to small or large cluster networks (Figure 4). There were, however, six 6+ clusters with members having T215D, M41L/T215E, D30N/N88D/M41L/T215C, M41L/L210W/T215D, M41L/T215S, and M41L/T215D (*n* = 6, 6, 7, 11, 11, and 11, respectively) (Figure 3). No 6+ clusters harbored M184V. Viruses bearing PIs and dual-class and multidrug resistance were rarer and largely singleton transmissions (Figure 3 and Figure 4). 

We evaluated the influence of select RAMs on HIV clustering patterns. Individual TDR surveillance mutations were sub-stratified based on their presence as isolated non-clustered transmissions or in association with small clusters (2–5 members) or large cluster networks (6+ members). The relative incidence of TDR RAMs according to cluster group category is summarized in Table 2.

In contrast, the prevalence of TDR to NNRTIs was influenced by clustering. K103N-bearing viruses were equally transmissible as singleton transmissions (4.2%), small (4.1%), or large clusters networks (4.3%) (Table 2). Episodic large 6+ clusters were implicated in onward spread of G190A (*n* = 160/160 members) and Y181C (9/9). There was a 6+ sub-cluster bearing the E138A natural polymorphism (*n* = 25 of 38 members) (Figure 3). 

We assessed the levels of pretreatment viremia in viral variants harboring M184V, TAMs, or NNRTI RAMs to compare their viral replicative fitness. As shown in Figure 5, median pretreatment viral loads among recently infected bearing M184V, TAMs, and NNRTI RAMs and no RAMs were 6811, 20,330, 38,847, and 46,088 copies/mL, respectively (one-way ANOVA, Kruskal Wallis statistic 197, *p* < 0.0001). Pretreatment viremia was significantly lower for variants bearing NNRTI RAMs associated with singleton transmissions as compared to small or large 6+ clusters having median viral loads of 19,952, 49,888, and 62,373 copies/mL, respectively (Kruskal Wallis statistic 14.91, *p* < 0.0001).

Taken together, K103N and G190A species showed no viral replicative fitness disadvantage. In contrast, M184V-bearing variants showed significantly lower viral loads (Figure 6). Similarly, baseline viral loads (mean log copies/mL ± SEM) in persons with non-B-subtype bearing M184V were 3.61 ± 0.14 log copies/mL, significantly lower than the baseline 4.36 ± 0.03, 4.28 ± 0.08, 4.12 ± 0.12 log copies/mL elicited by ART-naïve persons with non-B-subtype variants bearing NNRTI RAMs, TAMs, or no RAMs, respectively. To date, non-B-subtype infections are limited to small cluster networks (Figure 4). To date, large cluster non-B-subtype outbreaks do not bear resistant species.

### 3.4. Acquired and Transmitted Resistance to Integrase Inhibitor

There has been a widespread transition from NNRTI- to INSTI-based regimens. Second-generation integrase inhibitors, including DTG and BIC, show high potency and high genetic barriers to resistance. In Quebec, baseline genotyping across the integrase region is not universally recommended and is a small contributor to overall prevalence of TDR. To date, the prevalence of major mutations conferring resistance to first-generation integrase inhibitors, raltegravir, elvitegravir, and cabotegravir (CAB), was infrequent (under 3%) (Figure 6). The prevalence of ADR and TDR to DTG and BIC, e.g., R263K, is rare. The median (IQR) levels of viremia in individuals failing INSTI regimens without or with INSTI RAMs were 3935 (230–4295) and 2960 (498–25,159) copies/mL, compared to median 34,808 [16,17] copies/mL observed in INSTI-naive genotyped persons (KW statistic = 299, *p* < 0.001).

## 4. Discussion

ART is the mainstay for the lifelong management of HIV infections and population-level epidemic control. The optimization of simplified, tolerable, and durable ART regimens is predicated on a detailed understanding of time trends in the forward spread of HIV and drug resistance in MSM, HET, and recent migrants. The provincial drug resistance testing program in Quebec offered baseline genotyping for all newly infected persons in Quebec prior to ART initiation and following ART failure (viral loads > 200 copies/mL). Phylogenetic and clinical data characterized population-level time trends in the prevalence of TDR and ADR in people living with HIV (PLWH). 

We observed a high prevalence of TDR in B and non-B subtypes (14.7% and 13.9%, respectively). This included variants bearing NNRTIs (8.6–9.4%), TAMs (4.2–4.5%), and/or M184V (1.8–3.3%). Surprisingly, the incidence of transmitted RAMs to first generation NNRTIs, including K103N and G190A, and TAMs did not decrease over the 2017–2022 period as compared to the earlier 2012–2016 and 2007–2011 periods. This was observed despite the widespread transition to INSTI-based regimens (DTG, BIC, and CAB) and the discontinued use of EFV, AZT, and d4T.

Phylogenetic analyses revealed that the persistence of TDR to the NNRTIs was associated with small and large clustered outbreaks spreading among treatment-naïve populations. A large G190A cluster emerged as a micro-epidemic infecting 160 MSM over time. Multiple large cluster networks harboring K103N resistance have occurred, with the secondary acquisition over time of E138A- and P225H-resistant species in select clusters. The high incidence (4.3%) frequencies of transmitted K103N across singleton (4.2%), small (4.1%), and large cluster (4.3%) groups reflected its minimal fitness costs on viral replicative capacity. In contrast, the transmission of viruses having TAMs and T215 revertants arose as isolated singleton transmissions or small cluster groups. To date, domestic spread of TDR in persons harboring the non-B subtype has not occurred.

A recent cross-Canada study showed almost half of PLWH show suboptimal adherence to ART [18]. Of concern, an estimated 18% of persons show less than 85% adherence to oral ART. Despite the positive outcomes of available INSTI options, our studies emphasize the need to carefully consider pre-existing and archived species of NNRTI RAMs and M184V,I that may limit the long-term efficacy and durability of two-drug regimens, including CAB/RPV, DTG/RIL, or DTG/3TC [19,20,21,22,23,24]. 

The prevalence of M184V, which reduces 3TC and FTC susceptibility, has increased modestly from 0.8% in 2012–2016 to 3.3% in 2017–2022. The prevalence of M184V may be underestimated. Severe constraints of M184V on viral replicative fitness and rapid reversion dynamics (GTG→ATG) are observed within weeks or months in the absence of selective drug pressure. Notably, we observed that M184V was selectively associated with singleton transmissions, rather than small and large clustered networks. Although the low transmissibility of M184V is encouraging, M184V may have a potential impact on the future efficacy of PrEP and dual therapy options coupling DTG with 3TC or FTC. The rare acquisition and transmission of K65R, which reduces susceptibility to TDF and TAF, is promising. BIC and DTG three-drug options are forgiving, particularly for persons who show low adherence. 

The 2022 update of the International Antiviral Society-USA (IAS-USA) on drug resistance lists new mutations, including K101E/P, E138A/G/K/Q/R, V179L, Y181C/I/V, Y188L, G190E, H221Y, F227C, and M230I, that may reduce susceptibility to the second-generation NNRTIs, RIL, DOR, and/or ETV [25]. We observed that the E138A natural polymorphism was common, present in 3% of drug-naïve persons in Quebec. Notably, viruses bearing E138A were present as singleton transmissions and within small and large cluster networks. The E138A/K mutations have been observed in ATLAS and FLAIR trials in the rare cases of persons with virological failure on long-acting CAB/RPV injectable formulations [19,24,26]. A recent study shows the acquisition of NNRTI RAMs (K101E, K103R, E138K, V179D, Y181C, V189I, or M230L) in five persons failing long-acting CAB/RPV injectable therapy [19,24]. 

Taken together, TDR and archived resistance remain risk factors for virological failure [27,28]. Two-drug options, including CAB/RPV, DTG/3TC, and DTG/RPV, are a promising addition to the available treatment arsenal [8]. However, these regimens are recommended for virologically suppressed patients without treatment failure or suspected resistance. The Italian ARCA cohort showed only 44% of patients would be eligible to switch to CAB/RPV LA, after patients with a detectable viral load, presence of rilpivirine, and/or integrase strand transfer inhibitor mutations or a positive hepatitis B surface antigen result were excluded. Similarly, a French ART-naive cohort showed that 10% of patients had the HIV-1 A1/A6 subtype and rilpivirine- or cabotegravir-associated mutations that precluded the use of CAB/RPV [25]. The assessment of these conditions might only be feasible in many resource-rich settings, wherein baseline resistance testing is the norm and viral load monitoring is frequently performed.

## 5. Conclusions

Our studies applied a detailed molecular epidemiological approach to show the influence of transmission clustering in the spread of HIV-DR at a population level. Our findings highlight the added benefit of phylogenetics in understanding factors that govern HIV transmission and the spread of drug resistance and viral subtypes. Clustering facilitates the sustained spread of DR variants, particularly NNRTI RAMs, that persist over time and can affect disease course and limit options for antiretroviral therapy. Public health strategies that target early-stage infection, including rapid testing, PrEP, and routine genotyping may be of significant benefit in reducing the incidence and spread of DR species. Potent three-drug treatment options that favor selection of replicatively unfit variants, including M184V and K65R, may be forgiving in the setting of resistance or poor adherence. 

## Figures and Tables

**Figure 1 viruses-16-01230-f001:**
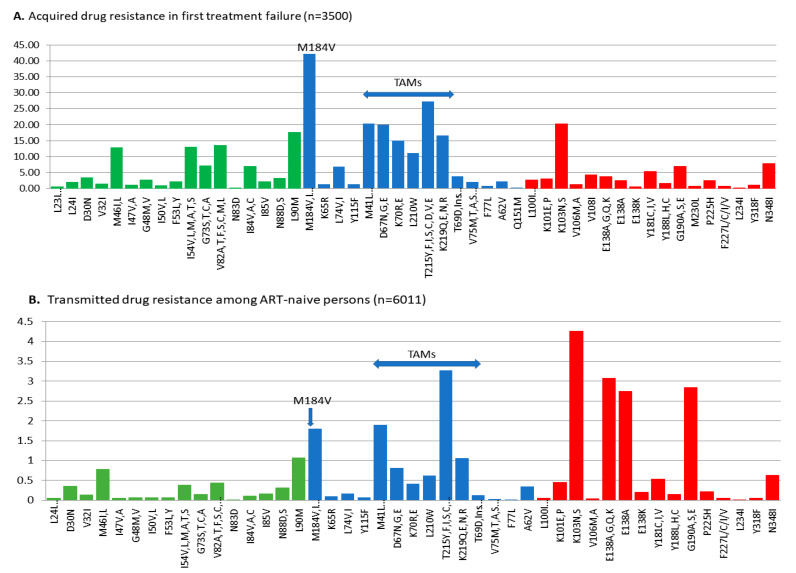
Frequencies of acquired and transmitted resistance-associated mutations (RAMs) in ART-experienced and ART-naïve groups with subtype B infections. (**A**) Rates of acquired resistance to PIs (green), NRTIs/TAMs (blue), and NNRTIs (red) among patients failing first ART treatment. (**B**) Rates of transmitted drug resistance mutations in newly genotyped, ART-naïve persons. Mutations are based on the updated World Health Organization 2009 list. Several mutations, including E138A, E138K, L234I, V318F, and N348I, associated with resistance to newer drugs rilpivirine and doravirine are also depicted.

**Figure 2 viruses-16-01230-f002:**
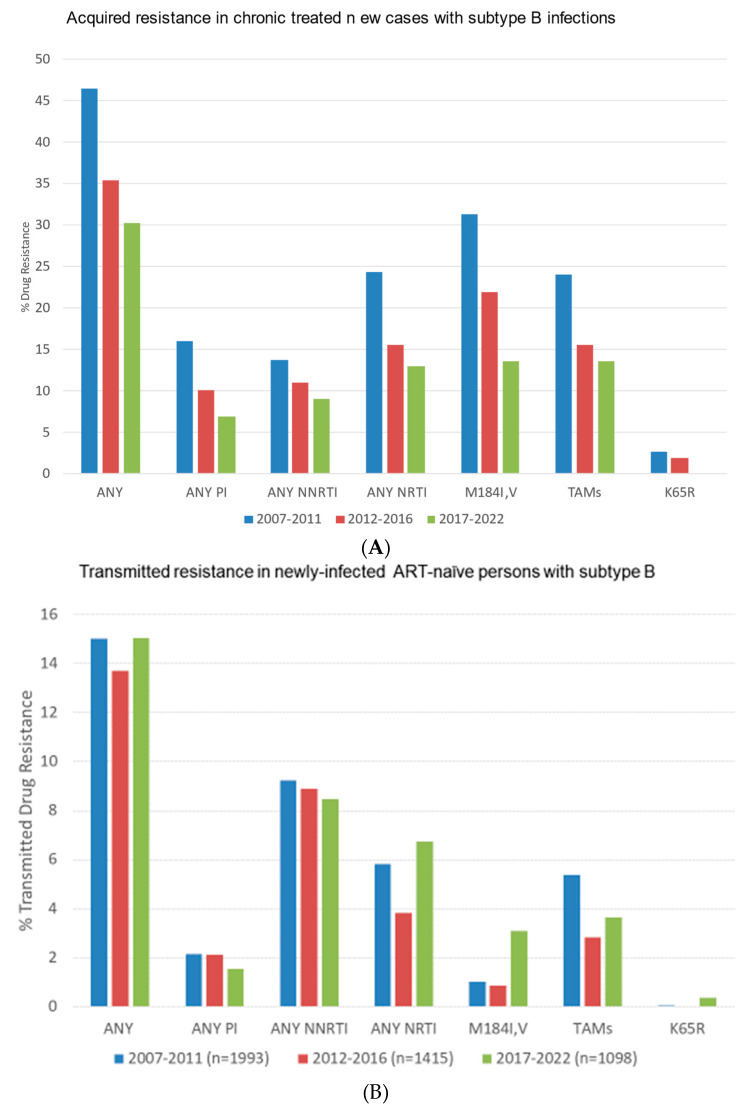
Time trends in the prevalence of acquired and transmitted drug resistance in first genotypes of persons bearing subtype B infections. (**A**). Frequencies of ADR in treated persons failing treatment (viral load > 100 copies/mL) in the 2007–2011 (blue), 2012–2016 (red) and 2017–2022 (green) periods. (**B**). Frequencies of TDR in treatment-naïve individuals during these three periods.

**Figure 3 viruses-16-01230-f003:**
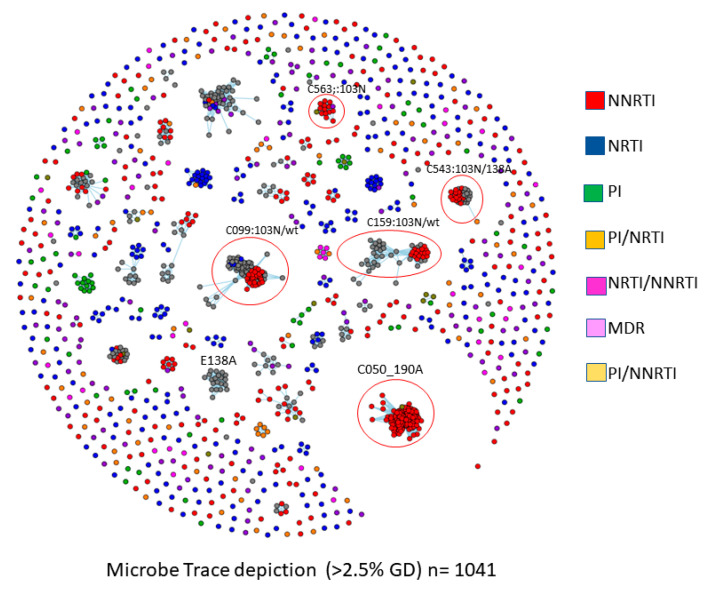
Influence of clustering on onward spread of TDR in treatment-naïve persons with subtype B infections. Representative large clusters bearing NNRTI RAMs (red nodes) are circled. Viral variants with resistance to NRTI mutations (blue nodes) include smaller clusters. Variants bearing PIs (green), and dual-class or multidrug resistance (MDR) are largely singleton transmissions.

**Figure 4 viruses-16-01230-f004:**
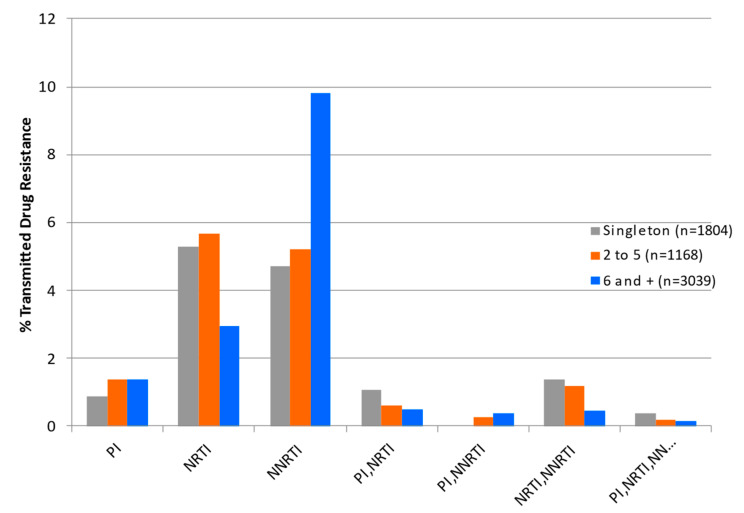
The impact of clustering on rates of transmission of resistance to nucleoside reverse transcriptase (RT) inhibitors (NRTIs), non-nucleoside RT inhibitors (NNRTIs), protease inhibitors (PIs), and dual- and triple-class multidrug resistance (MDR).

**Figure 5 viruses-16-01230-f005:**
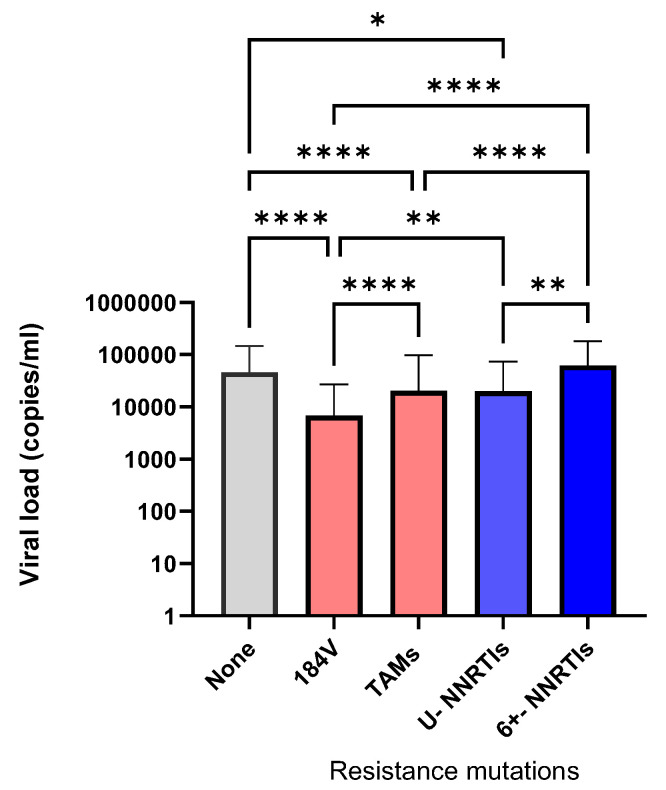
Influence of TDR mutations on viral replicative fitness. Pretreatment viremia (median ± IQR) in ART-naive persons with no RAMs (*n* = 5132), M184V/I (*n* = 258), thymidine analogue mutations (TAMs) (*n* = 359), and NNRTI RAMs present as unique (U) transmissions (*n* = 104) or within large 6+ clusters (*n* = 271, nonparametric ANOVA analysis and Dunn’s post hoc comparison tests of statistical differences in viremia between groups). *p*-values: * *p* < 0.05; ** *p* < 0.01; **** *p* < 0.0001.

**Figure 6 viruses-16-01230-f006:**
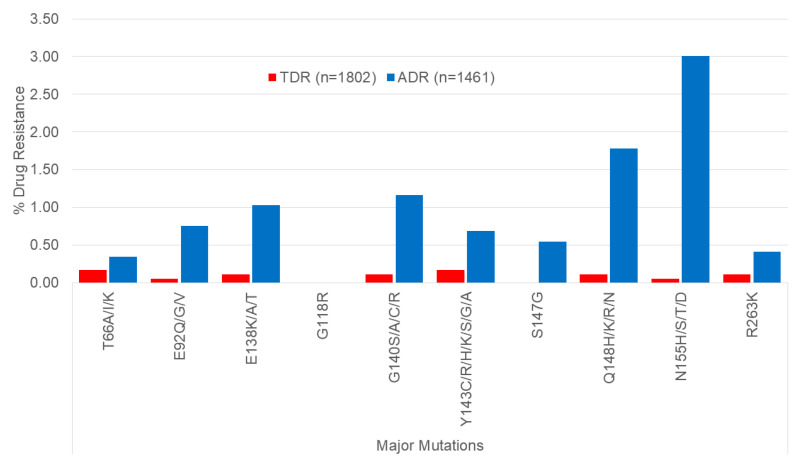
The prevalence of transmitted (TDR) and acquired (ADR) drug resistance among genotyped INSTI-naïve (*n* = 1802) and INSTI-experienced (*n* = 1461) persons in Quebec.

**Table 1 viruses-16-01230-t001:** Frequencies of acquired and transmitted drug resistance mutations (RAMs) in genotyped persons with subtype B and non-B-subtype infections.

Resistance-Associated Mutations	Acquired Drug Resistance (ADR) in ART-Experienced Persons (VL > 400 Copies/mL) ^a^			Transmitted Resistance in ART-Naïve Persons ^c^	
	Subtype B 1st ART Failure (*n* = 3500)	Subtype B CT Failures (*n* = 6013) ^b^	Non-B-Subtype 1st ART Failure (*n* = 488)	Subtype B TDR (*n* = 6011)	Non-B-Subtype PDR (*n* = 1572)
Any RAMs	58.5	71.2	36.7	14.7	13.9
PI RAMs	29.5	39.8	8.2	2.4	2.0
NRTI RAMs	38.1	62.0	27.5	5.9	6.2
M184V	42.2	50.0	23.6	1.8	3.3
TAMs	34.9	47.9	16.0	4.5	4.2
K65R	1.5	1.9	1.8	0.1	0.1
NNRTI RAMs	20.5	39.2	25.8	8.8	9.4
K103N	20.2	27.2	15.8	4.3	5.7
G190A	6.9	8.8	4.1	2.8	1.2
E138K	0.6	0.4	1.0	0.2	0.4
E138A	2.5	3.2	4.9	2.7	5.1

^a^ Acquired drug resistance in persons failing first ART regimen (VL > 400 copies mL). ^b^ Cumulative resistance (6013 sequences) in the 3500 treated persons (VL > 400 copies/mL). ^c^ Baseline transmitted drug resistance to newly infected, ART-naïve persons.

**Table 2 viruses-16-01230-t002:** The interrelationship between subtype-B-transmitted drug resistance and cluster group membership.

Drug Class RAMs	Specific RAMs	Cluster Group						All Subtype B (*n* = 6011)		Χ^2^ *p*-Value Exact Test Odds Ratio (Large Cluster vs. Singleton Transmission)	
		Singleton (*n* = 1804)		2–5 Members (*n* = 1168)		6+ Members (*n* = 3039)					
		N	%	N	%	N	%	N	%	*p* Value	Odds Ratio
Any		244	13.5	165	14.1	472	15.5	881	14.7	ns *	ns
NRTIs	Any NRTI	152	8.4	72	6.1	131	4.3	355	5.9	<0.0001	0.48 [0.39–0.62]
NRTI	M184I,V	60	3.3	19	1.6	29	1.0	108	1.8	<0.0001	0.28 [0.18–0.44]
TAMs	M41L	44	2.4	31	2.7	39	1.3	114	1.9	<0.001	0.52 [0.34–0.80]
	D67N,G,E	22	1.2	20	1.7	7	0.2	49	0.8	<0.0001	0.18 [0.08–0.42]
	K70R,E	19	1.1	2	0.2	4	0.3	25	0.4	<0.0001	0.13 [0.04–0.34]
	L210W	21	1.4	8	0.7	8	0.3	37	0.6	<0.001	0.22 [0.10–0.51]
	T215 revertants	77	4.3	47	4.0	73	2.4	197	3.2	<0.001	0.57 [0.40–0.76]
	K219Q,E,N,R	29	1.6	27	2.3	8	0.3	64	1.4	<0.0001	0.16 [4008–0.36]
NNRTIs	Any NNRTI	136	7.5	28	2.4	366	12.0	530	8.8	<0.0001	1.66 [1.35–2.04]
NNRTIs	K103N,S *	76	4.2	48	4.1	132	4.3	256	4.3	ns	ns
	G190A,S,E	8	0.4	10	0.9	153	5.0	171	2.8	<0.0001	8.70 [5.3–14.4]
	Y181C,I,V	14	0.8	7	0.6	11	0.4	32	0.5	ns	ns
	E138A,G,Q,K	45	2.5	46	3.9	94	3.1	185	3.1	ns	ns

All new infections in ART-naïve persons bearing subtype B infections were sub-stratified according to cluster group membership, i.e., singleton transmissions, small clusters with 2–5 members, or large clusters having 6+ members. The numbers and frequencies of resistance-associated mutations (RAMS) to NRTIs, thymidine analogues, and NNRTIs were determined for the three groups. Chi-square statistics ascertained the impact on cluster group association. The odds ratio (range) of being associated with a large cluster vs. singleton transmission is indicated. * ns, not significant.

## Data Availability

The Quebec HIV genotyping program sequences cannot be made publicly available for confidentiality reasons. An anonymized dataset of 233 sequences from PHI cohort participants belonging to the 30 largest transmission clusters is available on the Los Alamos HIV database website (GenBank accession numbers MK326905-MK327137). A representative portion of the tree has been previously published under GenBank accession numbers JF957375-JF957589. Analyzed HIV-DR datasets may be available from the corresponding author upon reasonable request.

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
