# Peer review of "Phylogenetic Network Analyses Reveal the Influence of Transmission Clustering on the Spread of HIV Drug Resistance in Quebec from 2002 to 2022"

_viruses, 2024, doi:10.3390/v16081230_

Round 1
Reviewer 1 Report
Comments and Suggestions for Authors
Brenner et al. presented findings from a phylogenetic analysis investigating the impact of transmission network on the spread of HIV drug resistance (HIV-DR) in Quebec, Canada. Leveraging the large database from the Quebec drug resistance program with sequencing data spanning 2002 to 2022, this study analyzed data from over 10,000 individuals, including both ART-naived and ART-experienced patients. The research team’s expertise in phylogenetic network analysis allowed them to identify the significant contribution of the transmission clusters to the spread of HIV-DR in this study. These findings underscore the critical need for targeted treatment and prevention efforts in high-risk population to curb the expansion of HIV-DR transmission cascades.
The study was well-designed, further boosted by its large samle size and rigourou data analysis methods, ensuing the robustness of tit findings. However, some issues remains to be addressed to improve and readability:
Major points:
1. Inconsistent RAM frequency numbers: There are many inconsistencies between the RAM frequencu numbers presented in the tables and the main text, which is misleading and confusing (detailed in “Minor points” section below). It is not appropriate to use approximate numbers in text while precise frequencies are presented in the tables, I think.
2. Missing Figure 4. Figure 4 is referenced in the text but is missing. Please verify the numbering of all the figures and ensure all referenced figures are included in the manuscript.
Minor points:
1. Line 61: Change “... reports in…” to “…report…”
2. Line 117~119: the numbering of the two gene fragments is confusing. Are they referring to different protocols used at different time periods ? For the 2262~ 3290 amplicon, it is impossible to cut it to a length of 1497 bp. Please correct or clarify.
3. Line 142: revise “The included...N348I. “ to form a full sentence.
4. Line 157~159: The descriptions here are misleading as the numbers differ from Table 1 (i.e. 8.6 ≠ 9, 2.4 ≠ 2). Corrections are needed .
5. Line 180: “…CT persons…” . The abbreviation ‘CT’ is not defiend anywhere in the text.
6. Line 186~187: Table 1 and Figure 1: Desciprions of E138A need to be consistent. If considered as assessory RAM, it should be included in Table 1 as well. If treated as a polymorphm only, it should not be listed alongside other RAMs in Figure 1.
7. TDR and ADR presentation: better to present TDR and ADR in the same order in Table 1 and Figures 1~2 for improved readability.
8. Line 242: show the full title all the data bars in the last group (PI, NRTI, NNRTI ?)
9. Table 2: The “All subtype B” percentages for “NRTIs” and “NNRTIs” differ from those in Table 1 (i.e. 5.9 ≠ 6.0, 8.8 ≠ 8.6).
10. Figure 6: Include definitions of U-NNRTI and 6+-NNRTIs in the legend.
11. Line 328 and 329: Numbers here are inconsistent with those in Table 1.
12. Line 339: The percentage for K103N is 4.3% in Table 1, not 4%.
13. High ratios of self-citations: Out of the 37 references, 15 cite the first-author’s previous work (12 as first author and 3 as co-author)…
Reviewer 2 Report
Comments and Suggestions for Authors
The work carried out to study the dynamics of HIV-1 drug resistance by analyzing molecular clusters is interesting and increases the existing knowledge about the features of the spread of viral resistance. The reviewer has no significant comments on the study design, analysis methods, results obtained and their interpretation. At the same time, there are a number of errors and inaccuracies that should be corrected to improve the quality of the manuscript.
1. Lines 157-158. The frequency of TDR is described by whole numbers. In this case, such rounding seems inappropriate. It is preferable to maintain the same accuracy (to tenths) as in other places in the text and Table 1.
2. In addition to groups, Table 1 also indicates individual mutations (M184V, K103N, K65R). However, the K65R mutation is far from the most common. These mutations should either be removed from the table or the reason for which they are listed separately should be explained.
3. It is recommended to redo Figure 1. The resolution does not allow reading individual mutations. The abbreviations DIS, MISC are not informative, and their explanation is missing in the text.
4. It is desirable to redo Figure 3, or indicate in the caption that N means NRTI, and NN means NNRTI.
5. Confusion with the numbering of figures. Lines 247, 249, 302 - Figure 4 is indicated, but it is missing from the manuscript. Line 269 - Figure 5 is indicated, but Figure 6 should be indicated.
6. Lines 385-386. The phrase should be clarified, for example, "... in Quebec", deleted, or extended. The approach to studying the speed of HIV DR spreading by analyzing the size of molecular clusters is already quite widespread, a series of works on this topic have been published in different countries.
